# Type-driven Neural Programming by Example

**Kiara Grouwstra**
Department of Computer Science
University of Amsterdam
Amsterdam, the Netherlands
`kiara.grouwstra@gmail.com`

**Emile van Krieken**
Department of Computer Science
Free University of Amsterdam
Amsterdam, the Netherlands
`e.van.krieken@vu.nl`

## Abstract

We propose a method to incorporate programming types into a neural program synthesis approach for programming by example (PBE). We introduce Typed Neuro-Symbolic Program Synthesis (TNSPS), and test it in a functional programming context to empirically verify whether type information helps to improve generalization in neural synthesizers on limited-size datasets. Our TNSPS model builds upon the existing Neuro-Symbolic Program Synthesis (NSPS) model [Parisotto et al., 2016], by incorporating information on types of input-output examples, of grammar production rules, as well as of the next node to expand in the program. Additionally, we introduce a generation method for programs written in a limited subset of the Haskell language. Our experiments show that incorporating type information using TNSPS improves the accuracy of the synthesized programs. This suggests that hybrid approaches that use both neural synthesis and strong type-checking is a fruitful research line.

Program synthesis is the task of automatically constructing a program that satisfies a given high-level specification [Gulwani et al., 2017]. Program synthesis is characterized by large search spaces, and has traditionally seen a split between formal versus neural approaches. Neural approaches (see Kant [2018] for an overview) typically use sequence-based deep learning models such as LSTMs [Hochreiter and Schmidhuber, 1997]. Formal approaches typically involve deductive techniques such as SAT solvers [Solar-Lezama and Bodik, 2008, Murali et al., 2017, Akiba et al., 2013, Alur et al., 2013, 2016, Torlak and Bodik, 2013, Zukoski and Wolpert, 2017] and types [Polikarpova et al., 2016, Osera, 2019, Brady, 2013]. If the synthesis specification consists of input-output examples, we get the field of programming by example (PBE). Type-theoretic program synthesis is based on types (as used in programming languages), typically with non-neural methods. [Polikarpova et al., 2016, Osera, 2019, Brady, 2013] Type-theoretic PBE uses types alongside examples [Osera and Zdancewic, 2015].

Input-output examples and types are complementary as specifications of program behavior: examples are expressive but can only evaluate complete programs while using only type as a specification is usually not descriptive enough, but provide a less noisy summary of program behavior. These differences imply the two may have synergy when combined. In this paper, we research whether neural program synthesis methods can benefit from using type information.

The main contributions of this paper are: **(a)** we bring the type-based information traditionally used in functional program synthesis to neural methods; **(b)** show that the neural synthesis of statically typed programs can benefit from techniques *specific* to these programming languages; **(d)** introduce a datatset generation method for programs in functional languages, including an open-source implementation.

34th Conference on Neural Information Processing Systems (NeurIPS 2020), Vancouver, Canada.

| task function | `let just = Just; compose = (.) in compose just unzip` |
|---|---|
| parameter types | `[(Int, Char)]` |
| output type | `Maybe ([Int], [Char])` |
| input expression | `([((17), '0'), ((20), '2')])` |
| output expression | `Right (Just ([17, 20], "02"))` |

Figure 1: A task function instance from our dataset with a corresponding sample input/output pair.

# 1 Neuro-symbolic program synthesis

First, we will introduce the *neuro-symbolic* program synthesis (NSPS) model introduced in Parisotto et al. [2016]. Our method extends NSPS because it is based on abstract syntax trees (ASTs) rather than unstructured sequences. This simplifies the process of incorporating type information. Assume that we have a domain specific language (DSL) formulated as a context free grammar with symbols $S$ (both terminal and nonterminal) and expansion rules $R$. NSPS uses a tree-based architecture called the *recursive-reverse-recursive neural network* (R3NN), which predicts rule expansions for *partial program trees* (*PPTs*). PPTs are ASTs that contain nonterminal leaf nodes. These nodes are called *holes* in the functional programming literature.

Every symbol $s \in S$ in the DSL has an $M$-dimensional representation, denoted $\phi(s) \in \mathbb{R}^M$, to each of which we concatenate the encoded input-output examples processed with a (bidirectional) LSTM. The R3NN then computes a representation for each node in the PPT in two steps. The first step is a recursive pass that uses a neural network $f_r : \mathbb{R}^{Q \times M} \to \mathbb{R}^M$ for each rule $r$, where $Q$ is the number of symbols on the RHS of $r$. We start with the representation $\phi(l)$ of the leaves of the PPT. Note that all leaves of a PPT are symbols (either terminal or nonterminal) and thus have a predefined embedding. Other nodes $n$ in the tree represent an expanded rule $R(n)$. We recursively compute $\phi(n)$ by applying $f_{R(n)}$ on its $Q$ children, up to $\phi(root)$ representing the full program. The second step is the reverse-recursive pass that again uses a neural network $f_g : \mathbb{R}^M \to \mathbb{R}^{Q \times M}$ for each rule $r$. Like the recursive pass, it recursively computes node representations $\phi'(n)$. However, it does this from the root towards the leaves, so in reverse order. They then process these embeddings with a bidirectional LSTM.

Like for symbols, NSPS also has an $M$-dimensional representation $\omega(r) \in \mathbb{R}$ for each rule $r$. A possible *expansion* $e \in E$ of a hole $e.l$ is a grammar production rule $e.r$. The score of an expansion $e$ is defined as the dot product of their respective embeddings: $z_e = \phi'(e.l) \cdot \omega(e.r)$. This model uses a cross-entropy loss and strong supervision, i.e. it supervises our training on the actual task function.

# 2 Methodology

Here we will briefly discuss our synthesis DSL, then explain our PBE model, which applies type information to improve synthesis quality and how it is incorporated in the NSPS model Parisotto et al. [2016]. To test our hypothesis we generate a PBE dataset in the functional programming domain, implemented by using a subset of Haskell [Jones, 2003] as our synthesis DSL. Viewing a program as a composition of function applications guarantees us that any complete program filtered to the right type from the root that also passes a compiler type-check will yield us output of the desired type. This helps us reduce our synthesis search space to a sensible subset, devoid of e.g. programs containing variable definitions that end up never being used. This guarantees that our search will focus on finding acceptable solutions. An example showing what different components of our dataset items might look like may be found in Figure 1. We *unroll* any function applications in our grammar, such that given a binary function `and` and a symbol `false` taking no arguments it might look as follows:

```
expr = "(and ", expr, " ", expr, ")";
expr = "(and ", expr, ")";
expr = "and";
expr = "false";
```

### 2.1 Typed Neuro-Symbolic Program Synthesis

Here we describe how we adapt the NSPS model to our domain, as well as how we further augment the model by type info as neural features.

We will now explain how we augment the NSPS model to incorporate type info. Consistent with how we embed expressions, we similarly stringify types, then one-hot encode their characters as we do for input/output expressions. To get the most out of our types, we will want to provide them for: **(a)** inputs and outputs, which we simply incorporate as additional features in Parisotto et al. [2016]'s *example encoder* as mentioned in Section 1, concatenating their one-hot embeddings to those of the input/output pairs before passing them through the input/output LSTMs, making for a total of $8HT$ features per sample; **(b)** expressions from expansion rules $r$; for these we may calculate types statically upfront, then embed these to obtain $M \cdot T$ features per expansion rule $r \in R$, and during R3NN prediction concatenate these features to the existing representation $\omega(r) \in \mathbb{R}^M$, yielding $\omega'(r) \in \mathbb{R}^{M \cdot (T+1)}$. **(c)** (hole) AST nodes $c$ in any PPT. During prediction in the R3NN, we embed these types by an LSTM into $M \cdot T$ features per hole type. As with rule embeddings, we then concatenate these with the original $M$ hole node features, yielding $\phi''(l) \in \mathbb{R}^{M \cdot (T+1)}$.

Having obtained our respective rule and hole embeddings expanded to $M \cdot (T + 1)$ from the original $M$ features, we would then calculate the scores from these enhanced embeddings using the same calculations as before, swapping out the embeddings to their enhanced versions, i.e. from $z_e = \phi'(e.l) \cdot \omega(e.r)$ to $z_e = \phi''(e.l) \cdot \omega'(e.r)$.

## 3 Experimental setup

Our experiment aims to test if our type-augmented model adds value over the baseline model, as per our hypothesis. We evaluate our models on a dataset we generate, using 4 runs for each model. Our models include a uniform random synthesizer, our vanilla implementation of Parisotto et al. [2016]'s NSPS model, TNSPS (keeping $H{=}32$ but allotting that same amount for types), as well as an enlarged version of the vanilla model doubling the features per i/o sample to match TNSPS for fair comparison. We first find a learning rate appropriate for our experiment on our vanilla implementation of NSPS, otherwise taking the hyperparameter values described in Section A.

To generate a dataset we pick our own set of types and operators as described in Section B.3. For this we have picked a limited set of operators widely applicable over the types used. Types used include `Char`, `Int`, `Maybe`, `List`, `(,)`, and `Either`. We have limited our DSL to the following operators for our chosen types: `0`, `Just`, `maybe`, `(:)`, `length`, `(,)`, `zip`, `unzip`, `toEnum`, `fromEnum`, `foldMap`, `elem`, `sequenceA`, `sequence`, `fmap`, `mempty`, `(<>)`, and `(.)`. Our dataset is limited to programs of up to 3 nodes, each containing the symbol of one such operator.

Translating Parisotto et al. [2016]'s synthesizer from its original *FlashFill* [Gulwani, 2011] domain to our domain of *functional programs*, we made the following adjustments: **(a)** Input-output samples used by Parisotto et al. [2016] were all strings. Samples in our functional domain can be arbitrary expressions. We simply stringify these, then one-hot encode the strings' characters as Parisotto et al. [2016] did using their string samples. **(b)** We aggregate losses over an epoch by taking their mean.

## 4 Result

Any results here are trained on our dataset of programs of up to 3 nodes, during training evaluated as in Parisotto et al. [2016] by sampling 100 programs from the synthesizer for any task function instance. Accuracy results during and after training may be found in Figures 2a and 3.

We see that vanilla NSPS learns to a more limited extent before converging. Our task seems relatively challenging, with accuracy for the baseline model increasing only somewhat beyond its initial random accuracy. Furthermore, most of the gains in accuracy for the baseline model are attained in the initial 10 epochs of training. These issues may be largely due to the limited size of our dataset.

Our enlarged model fared little better than the baseline, again likely due to generalization issues related to the size of our dataset. Our 'typed' NSPS model however starts from sub-random accuracies, yet ends up able to learn more, after 20 epochs out-performing both our baseline and enlarged models. This indicates it is in fact worthwhile to distribute features between input/output pairs and types.

**(a)**

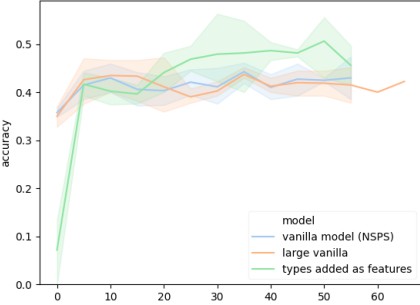

**(b)**

| p-values | vanilla | large | types | uniform |
|----------|---------|-------|-------|---------|
| **vanilla** | 1.000 | 0.859 | 0.010 | 0.013 |
| **large** | 0.859 | 1.000 | 0.024 | 0.065 |
| **types** | 0.010 | 0.024 | 1.000 | 0.001 |
| **uniform** | 0.013 | 0.065 | 0.001 | 1.000 |

Figure 2: **(a)** Validation accuracy over 100 samples across training epochs for our models **(b)** P-values of acuracy on 100 samples between our models

| | evaluated @ 20 samples | | | | | evaluated @ 100 samples | | | | |
|---|---|---|---|---|---|---|---|---|---|---|
| | accuracy | | acc mean @ x nodes | | | accuracy | | acc mean @ x nodes | | |
| model | mean | var | 1 | 2 | 3 | mean | var | 1 | 2 | 3 |
| NSPS | 0.13 | 0.000 | 0.27 | 0.13 | 0.11 | 0.37 | 0.000 | 0.77 | 0.36 | 0.30 |
| large | 0.12 | 0.001 | 0.30 | 0.12 | 0.09 | 0.38 | 0.002 | 0.73 | 0.38 | 0.30 |
| typed | **0.22** | 0.002 | **0.55** | **0.24** | **0.12** | **0.48** | 0.003 | 0.77 | **0.55** | **0.34** |
| random | 0.14 | 0.000 | 0.43 | 0.11 | 0.11 | 0.33 | 0.000 | **0.84** | 0.25 | 0.31 |

Figure 3: Summary of test set accuracy on our test set over different models after training (4 seeds each), for each selecting the best-performing epoch by early stopping.

As the 3-node programs turned out fairly hard, we cannot yet distinguish a clear correlation between task function node size and advantage from type information. We compare the accuracy across our models by an independent two-sample t-test, see Figure 2b. This shows only a p=1.0% chance our typed model stemmed from the same distribution as the baseline, showing the improvement is statistically significant.

## 5 Discussion

### 5.1 Design limitations

We disregard any programs exceeding a limit of 3 nodes, both on dataset generation as well as on synthesis. Finally, our implementation unfortunately still uses *local types* from our unrolled grammar over full type inference, so it cannot use type information from elsewhere in the program tree.

### 5.2 Topics for future research

As neural synthesis methods aimed at PBE in the functional programming domain is a broad topic encompassing a variety of design decisions, we have had to leave some questions unanswered. First, we might *pre-compile* partial programs after each synthesis step to provide a synthesizer using weak supervision (reinforcement learning) with an immediate reward signal on whether a program type-checks. While our present experiment used types based on input/output examples, another question is if it could help to use the true type signature of the task function.

### 5.3 Conclusion

We presented a method to incorporate programming types into a neural program synthesis approach for programming by example. We generated a dataset in the functional programming context, and demonstrated type information to improve synthesis accuracy even given a comparable number of parameters. Finally, we suggest a number of topics of interest for future research in type-driven neural programming by example.

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

## Appendix

## A  Hyperparameters

### A.1  Hyperparameters used for dataset generation

In this section we will describe the hyperparameter values we have used in our dataset generation.

We generate types to substitute into *type variables* using a maximum of only *one* level of nesting, i.e. allowing type *list of booleans* though not type *list of lists of booleans*.

For any parameter type containing type variables used in task functions, we generate a maximum of 5 type *instances*, before deduplication. Whereas Parisotto et al. [2016] generated 10 inputs for each task function, we instead generate up to 10 for each *type instance* of a task function, before deduplicating.

While they limited functions to a maximum of 13 operations, we instead limit ours to a maximum of 3, given that our current operator set is considerably bigger than those of their *FlashFill* domain.

Numbers that we generate, all of them integers, we limit to the range from $-20$ to $20$. For characters we stick to the range of digits, i.e. from '0' to '9', a decision made with the intent to let their characters overlap with those of digits for the purpose of helping reduce characters used in the encoder, in turn reducing the size of its one-hot embedding. This arbitrary constraint serves no other purpose than to constrain required compute.

Data structures such as *string*, *list*, *set*, and *hashmap*, we each generate using lengths in the range from 0 to 5. Of these, *sets* might further deduplicate down, as this structure only holds unique items.

Our dataset we split into training, validation and test sets using a ratio of $35\%$, $35\%$, and $30\%$, respectively. As [Parisotto et al., 2016] we sample $1,000$ training programs from the total function space.

### A.2  Hyperparameters in our synthesizer

In this section we will describe the hyperparameter values we have during the training and evaluation of our synthesizers.

We use 3 layers in our LSTMs, which are present in our sample encoder (for both input and output), our type encoders (for rule expansions and holes), as well as for sample conditioning and scoring in our *R3NN*. We do allow bias terms although the original paper did not show these in their formulas. We train for a maximum of $1,000$ epochs.

Our *encoders* process items (either input-output samples or types) using a batch size of $8$. Our *R3NN* must use a fixed number of embedded input-output pairs on the basis of its LSTM used for conditioning, and as such we have fixed this to use samples of $8$ embedded input-output pairs.

As Parisotto et al. [2016], for synthesizer evaluation we sample $100$ functions from the model for each task function, determining success based on the best from this sample, i.e. considering the synthesis a success if any of these pass our PBE task, exhibiting the desired behavior.

We evaluate performance on our validation set once after every 5 epochs of training. During evaluation we similarly check for convergence based on the loss, averaging over windows of 2 evaluations, i.e. stop training if the validation loss over the past two evaluations has increased from the two before.

We arbitrarily limit synthesized functions to the same complexity limit of 6 operators as used during generation of task functions. We allow 32 features in our symbol and expansion rule embeddings, i.e.

$M$ in Parisotto et al. [2016]'s *R3NN*. We allow 32 features per input or output per LSTM direction, i.e. $H$ in Parisotto et al. [2016]'s *sample encoder*. We clip gradients to stay within a range from $-1$ to $1$. The learning rate for our Adam optimizer we search over by a grid search using our vanilla NSPS model, considering values of $1e-2$, $1e-3$, $1e-4$, and $1e-5$. Of these, we settle on a learning rate of $1e-2$.

## B    Miscellaneous experiments

Aside from our main experiment, we also tried a few other configurations for which we had not managed to obtain conclusive results.

### B.1    Type filter

The first of these was the idea to combine a synthesizer with a compiler check to filter out any non-compiling programs. While the downside to this would be that the synthesizer would be made dependent on this extra compiler check, incurring a run-time penalty during synthesis, linear in the number of expansion rules provided, the advantage to such a setup would be that the synthesizer would no longer need to learn to disregard non-compiling programs itself, reducing synthesis to a ranking problem of the compiling (partial) candidate programs. We achieve this by simply masking the predicted scores of uncompiling programs in our NSPS implementation (before calculating actual probabilities by softmax) to have no probability, i.e. $p(e) = 0.0$.

While we failed to obtain any significant improvement over the baseline model using this setup, this result may well have related to our implementation. We presently used the `hint` Haskell library as our interpreter for type-checks, which unfortunately yielded false positive compiler errors for types containing ambiguous type variables, such as `show undefined`, which the Haskell compiler would resolve to type *string*, whereas the `hint` library would complain that the `undefined` argument would prevent resolving `show`'s type variable.

As this counter-factual signal would prevent this synthesizer from correctly synthesizing the affected programs, the fact that it nevertheless performed on par with our baseline algorithm suggests this approach does in fact have potential. While we might have addressed this flaw in our implementation by switching from this interpreter library to using Haskell's compiler API directly, due to time constraints this unfortunately fell out of scope for this thesis.

### B.2    Picking holes

Although the topic of which hole to fill was not directly touched upon in Parisotto et al. [2016], our baseline implementation had the synthesizer deterministically fill the first hole (under any given order — we used left-to-right). Nevertheless, we did also wonder what the effect might be if we would allow filling any hole.

During training, we would then opt to randomly pick a hole to try and fill. On evaluation, we would then look at the confidence scores for any hole expansions across holes, sampling from this full matrix rather than just the vector slice corresponding to the first hole. This allows the synthesizer to take into account the relative confidence of expansions for different holes, enabling it to forego holes involving more uncertainty in favor of those it feels more confident about, which may in turn provide additional information that may then reduce ambiguity for the remaining holes. [1]

Unfortunately, we obtained inconsistent results on this model versus our baselines across different experiment attempts, originally getting the expected improvement, although in our final implementation we had not managed to reproduce this improvement. We had to leave further analysis of these inconsistent results out of scope due to time constraints, and as such feel hard-pressed to make definitive statements on the effectiveness of this approach. Nevertheless, we consider this to be a topic of interest in AST-based neural program synthesis.

---

[1] An additional advantage of this would be it could more uniformly explore various partial program trees across synthesis steps. That said, uniform exploration there isn't necessarily the ideal situation — one might for example imagine using weights to prioritize situations our synthesizer is less confident about.

## B.3 Dataset generation

As we were unable to find existing datasets in the functional program synthesis domain of a size appropriate for training a neural model, we have opted to instead generate a dataset of our own. As the potential space of viable programs is potentially unbounded, we instead opt to artifically limit the space to generate from.

Our main goal in creating a dataset consists of generating the programs to be synthesized, alongside the input-output data we would like to use to synthesize them from (as per our PBE setting). Now, the inputs here are generated, whereas the outputs are obtained simply by running these inputs through our programs.

However, as our programs may take parameters of parametric types, e.g. list of any given type `[a]`, we take the intermediate step of instantiating such types to monomorphic types, i.e. types not containing type variables themselves, which we may then generate inputs for.

Note that to make our task easier, we further maintain such a separation by type instances for our generated programs, meaning that a potential *identity function* in our dataset might be included in our training set under type instance $Int \rightarrow Int$, then perhaps in our test set under another type instance like $Char \rightarrow Char$. We may sometimes still refer to just task functions however, as the distinction is not otherwise relevant.

An example showing what different components of our dataset items might look like may be found in Figure 1.

Our full generated dataset consists of the following elements:

- the right-hand symbols or operators we allow in our DSL, to be detailed in Section 3;
- the types of any task function in our dataset;
- sample input-output pairs for different type instances of our task functions;
- a split over training/validation/test sets of any of our tasks, i.e. type instances for a given task function;
- pairs of symbols in our DSL with their corresponding expansion rules (including type annotations for holes);
- types of any expansion rules in our DSL;
- NSPS's maximum string length $T$, based on our stringified input-output examples (also taking into account types for the augmented model);
- mappings of characters to contiguous integers so we can construct one-hot encodings covering the minimum required range of characters (tracked separately for input-output, types, and either);
- the configuration used for data generation to make data reproducible, discussed further in Appendix section A.1;
- the types we generate to instantiate type variables, again for reproducibility purposes, separated by arity based on the number of type parameters they take.

A brief overview of how to generate such a dataset to train our synthesizer on is shown in Algorithm 1.

We first generate our expansion rules by *unrolling* each operator in the dataset as illustrated in Section **??**, using a different number of holes corresponding to any applicable arity.

To create our dataset of task functions, we start from an expression consisting of only a hole, then step by step generate any type-checking permutation by filling a hole in such an expression using our expansion rules. We only fill holes in a generated expression up to a user-defined limit, disregarding any programs still containing holes after this point.

Like Parisotto et al. [2016] we uniformly sample programs from our DSL, based on a user-defined maximum, while still respecting the above complexity limits. We similarly use sampling for the generation of sample input-output pairs and, for instantiating our type variables, monomorphic types, i.e. types not containing type variables.

---

**Algorithm 1** dataset generation

---

**given**: expression space $E$, operators or symbols $s \in S \subset E$, expansion rules $r_s \in R \subset E$, programs $p \in E$, types $t \in T$, monomorphic types $t^{(m)} \in T^{(m)} \subset T$, input expressions $i \in E$, output expressions $o \in E$, parameters $a$;

**calculate** expansion rules $r_s^{(1,\dots,n)}$ **from** $s \in S$ **by** unrolling our grammar symbols;

**generate** any possible program $p$ **given** expansion rules $\forall s : r_s^{(1,\dots,n)} \in R^n$ and a max number of holes;

**sample** monomorphic types $t^{(m)} \in T^{(m)}$ **up to** a max number and **within** a given nesting limit;

**generate** instances $t^{(m)}_{a_p^{(1,\dots,n)}}$ **for each** generic non-function parameter types $\forall p : t_{a_p^{(1,\dots,n)}}$ **given** sampled types $t^{(m)}$;

**sample** type instances $t_p^{(m)}$ **for each** function type $\forall p \in E : t_p$ **up to** a given number;

**generate** sample expressions $i^{(1,\dots,n)}_{t^{(m)}_{a_p^{(1,\dots,n)}}}$ **for each** non-function parameter type instance $t^{(m)}_{a_p^{(1,\dots,n)}}$, **up to** a maximum each and **within** given value bounds;

**calculate** a filtered map of generated programs $p^{(1,\dots,n)} \in E$ **for each** instantiated function parameter type combination $\forall a_p : t^{(m)}_{a_p^{(1,\dots,n)}}$ **by** matching its type to obtain samples $i^{(1,\dots,n)}_{t^{(m)}_{a_p^{(1,\dots,n)}}}$ for our function types;

**calculate** outputs $o^{(1,\dots,n)}_{t_p^{(m)}}$ **for each** task function instance $t_p^{(m)}$ **given** a sample of generated inputs $i^{(1,\dots,n)}_{t^{(m)}}$;

**filter** out program type instances $t_p^{(m)}$ **without** i/o samples $(i, o)^{(1,\dots,n)}_{t_p^{(m)}}$;

**filter** out any functions instances $t_p^{(m)}$ **with** i/o behavior identical to others to prevent data leakage;

**sample** task function type instances $t_p^{(m)}$ **from** any remaining programs $p$;

**calculate** longest strings and character maps;

**split** our task function type instances $t_p^{(m)}$ **over** train, validation and test datasets.

---

While we quickly mentioned type-checking programs to filter out bad ones, we had yet to expand on this practice: we presently use a Haskell interpreter to type-check our generated programs at run-time, filter out non-function programs (e.g. `false`), and check if program types look *sane*: to weed out some programs we deem less commonly useful, we filter out types *containing* functions (e.g. list of functions), as well as types with constraints that span more than a single type variable (e.g. $(Eq(a \rightarrow Bool)) \Rightarrow a$). [2]

As we cannot directly generate samples for types containing type variables, we first instantiate any such type variables using a fixed number of monomorphic types we generate. We define a maximum level of type nesting for such sampled types, to prevent generating types like 'list of lists of booleans'. We further specify a maximum number of types generated.

We then use these monomorphic types to instantiate any polymorphic (non-function) input types occurring in our task functions. To simplify things, we restrict ourselves to substituting only non-parametric types (e.g. boolean yet not list of boolean) for type variables contained in a larger type expression. In the event the type variables in our types involve *type constraints*, we ensure to only instantiate such type variables using our monomorphic types that satisfy the applicable type constraints.

This yields us a set of monomorphic input types, for which we then generate up to a given maximum number of sample inputs, although this may get less after filtering out duplicate samples. We use hyperparameters to indicate range restrictions for different types here.

---

[2]Programs not passing these checks are not necessarily invalid, but by our engineering judgement, are much more circumstantial in their usage, making for only a smaller portion of valid programs, aggravating our search space problem. For this reason, we would currently prefer for our synthesizer to focus on the region of our search space that we generally deem to be of higher interest.

For any given given task function type signature, we then check for the types of each of their input parameters, and take any corresponding combination of type instances in case of polymorphic types.

Now, for any non-function parameter types, we may just take the previously generated sample input-output pairs for those types. Parameters with function types, however, we instead instantiate to function values by just taking any of our generated task functions corresponding to that type.

Based on these sample inputs, we would then like to generate corresponding outputs for our generated task functions. For our task functions that are polymorphic, i.e. contain type variables, we must do this separately for different type instances.

We run our programs using our run-time Haskell interpreter. We catch run-time errors on specific inputs such that we can regard these errors as just another resulting output that our synthesizer should consider when comparing behavior between programs. In other words, a *partial function*, i.e. a function that only works on a subset all inputs of the desired input types, may still constitute a valid program that we may wish to learn to synthesize.

Having generated input/output examples for our task functions, we finally filter out any task function type instances for which we have somehow failed to generate such samples. We moreover limit our dataset to a given maximum.

At this point we:

- use a random split to divide our task function type instances over training, validation and test sets;
- calculate the longest input-output examples in our dataset (as string), when considering types (as per our experiment) also taking into account the length of the string representations of such types of inputs and outputs;
- track any characters used in string representations of the expressions in our dataset (for our type experiment also those used in string representations of the types), and assign them to indices for our *one-hot* encodings of input-output examples (and their associated types).

To prevent data leakage, we ensure no task function instances across different datasets share the same *input-output pairs*. When deciding which task function instance of a similar pair to keep, we first look for the more general function (i.e. operating across more type intances as used in our dataset), otherwise look for the task function with the shortest implementation (in terms of number of nodes), or finally, as a tiebreaker, arbitrarily keep either of the two.

