# OpenReview forum: "Type-driven Neural Programming by Example"
_NeurIPS.cc/2020/Workshop/CAP — NeurIPS 2020 CAP Workshop_

### Official Review · AnonReviewer1 · 2020-10-30
**Type-driven Neural Programming by Example**

**Rating:** 7
**Confidence:** 4

**Review:**

This work proposes a program synthesis model, Typed Neuro-Symbolic Program Synthesis, for solving programming by example problems in a functional programming domain.

The paper examines an important question, namely, whether incorporating type information into neural program synthesis systems leads to better performance, particularly in the low data regime. To that end, this paper compares the neuro-symbolic program synthesis (NSPS) model introduced in Parisotto et al. [2016] to a version which encodes type information.

Results indicate that type information does help synthesis performance.

In general, the questions studied here are valuable and relevant to the neural program synthesis community.

Comments:
- It would be very interesting to study how the performance difference between TNSPS and baselines varies as the training dataset size varies.
- Likewise, does incorporating type information help generalization to out-of-distribution programs (for example, programs longer than those seen during training? See https://arxiv.org/abs/1611.01989, https://arxiv.org/pdf/1902.06349.pdf, and https://arxiv.org/pdf/1809.04682.pdf for experiments in the list processing domain varying program size.)

---

### Decision · Program_Chairs · 2020-11-02

**Decision:**

Accept

**Comment:**

I recommend acceptance here, as the review is positive.